



# Nitrous oxide variability at sub-kilometre resolution in the Atlantic sector of the Southern Ocean

Imke Grefe[1,*], Sophie Fielding[2], Karen J. Heywood[1], Jan Kaiser[1]

[1]Centre for Ocean and Atmospheric Sciences, School of Environmental Sciences, University of East Anglia, NR4 7TJ Norwich, UK
[2] British Antarctic Survey, High Cross, Madingley Road, Cambridge, CB3 0ET, UK
[*]Now at: Lancaster Environment Centre, Lancaster University, LA1 4YQ Lancaster, UK

*Correspondence to*: Imke Grefe (i.grefe@lancaster.ac.uk), Jan Kaiser (j.kaiser@uea.ac.uk)

## Abstract

The Southern Ocean is an important region for global nitrous oxide ($N_2O$) cycling. The contribution of different source and sink mechanisms is, however, not very well constrained due to a scarcity of seawater data from the area. Here we present high-resolution surface $N_2O$ measurements from the Atlantic sector of the Southern Ocean, taking advantage of a relatively new underway setup allowing for collection of data during transit across mesoscale features such as frontal systems and eddies. Covering a range of different environments and biogeochemical settings, $N_2O$ saturations and sea-to-air flux were highly variable: Saturations ranged from 96.5 % at the sea ice edge in the Weddell Sea to 126.1 % across the Polar Frontal Zone during transit to South Georgia. Negative sea-to-air fluxes of up to -1.3 µmol m$^{-2}$ d$^{-1}$ were observed in the Subantarctic Zone and highest positive fluxes of 14.5 µmol m$^{-2}$ d$^{-1}$ in Stromness Bay, coastal South Georgia.

## 1 Introduction

Nitrous oxide ($N_2O$) is a strong greenhouse gas and currently the third largest contributor to radiative forcing after carbon dioxide ($CO_2$) and methane ($CH_4$) (Hartmann et al., 2013). Furthermore, $N_2O$ is an important source for stratospheric $NO_x$, which is involved in catalytic ozone depletion (Crutzen 1970; Ravishankara et al. 2009). The ocean, including coastal zones, estuaries and rivers, is estimated to contribute approximately 25 % to global $N_2O$ emissions (Myhre et al., 2013). The Southern Ocean alone is estimated to account for 5 % of global emissions (0.9 Tg a$^{-1}$ N (nitrogen equivalents), Nevison et al., 2005). However, measurements of oceanic $N_2O$ concentrations in this region are scarce and these emission estimates are based on atmospheric measurements at Cape Grim and a few seawater measurements in the Southern Ocean from 1977-1993 (Nevison et al., 1995, 2004 and references therein).

The Scotia Sea in the Atlantic sector of the Southern Ocean is confined by the Scotia Ridge to the north, east and south and the Drake Passage to the west (Atkinson et al., 2001). South Georgia is part of the North Scotia Ridge in the path of the Antarctic Circumpolar Current (ACC) and within the Antarctic Zone (AAZ) south of the Polar Front (PF). Waters around South Georgia are characterised by high abundances of phytoplankton, zooplankton and vertebrate predators (Atkinson et al.,



2001), whereas most other areas of the Southern Ocean are dominated by High Nutrient Low Chlorophyll (HNLC) conditions (Martin, 1990). A relatively stable water column and benthic iron input support productivity in the vicinity of the island (Holeton et al., 2005; Korb et al., 2005).

To the south of the Scotia Sea the Weddell Sea is characterised by a large gyre, flowing eastwards until 20-30° E, returning
westwards along the continental margin and following the eastern coast of the Antarctic Peninsula northwards (Deacon, 1979). Seasonal blooms in the Weddell Sea are associated with the Antarctic shelf and the ice edge (El-Sayed and Taguchi, 1981; Kristiansen et al., 1992; Nelson et al., 1989; Smith Jr and Nelson, 1990). Furthermore, drifting icebergs stimulate productivity by input of terrigenous iron through melt water (Biddle et al., 2015; Smith et al., 2007).

Generally, the Southern Ocean has the potential for both production and removal of $N_2O$ (Rees et al., 1997): Solubility of
$N_2O$ increases at lower temperatures, and together with downwelling areas associated with deep-water formation and convergences in the Antarctic frontal zones, wide areas could function as sinks. On the other hand, upwelling of deep and intermediate waters could be a source of biologically produced $N_2O$ to the atmosphere. Artificial iron fertilisation of the Southern Ocean may stimulate biological $CO_2$ uptake, but could potentially increase $N_2O$ production, offsetting the benefits of $CO_2$ sequestration in terms of radiative forcing (Fuhrman and Capone, 1991; Jin and Gruber, 2003). While an iron
fertilisation experiment in the Australasian sector of the Southern Ocean showed $N_2O$ accumulation in the pycnocline (Law and Ling, 2001), no increase in $N_2O$ concentrations was observed during a similar experiment in the Atlantic subpolar sector (Walter et al., 2005).

Here, we present high-resolution measurements of ocean surface $N_2O$ concentrations from the Scotia Sea and Weddell Sea. Using off-axis Integrated Cavity Output Spectroscopy (ICOS) (Arevalo-Martinez et al., 2013; Grefe and Kaiser, 2014)
combined with wind-speed gas exchange parameterisations we can resolve small-scale variability in $N_2O$ fluxes and capture the impact of frontal structures and changes in weather conditions on emissions from the Southern Ocean.

## 2 Methods

$N_2O$ concentrations in surface waters were measured during the annual Western Core Box (WCB) krill survey in the Scotia Sea between 28 December 2011 and 16 January 2012 (JR260B) and in the Weddell Sea from 20 January to 2 February 2012
(JR255A/GENTOO Gliders: Excellent New Tools for Observing the Ocean) on board RRS *James Clark Ross*. The measurement region for both cruises is shown in Fig. 1.

The setup and performance of the coupled analyser-equilibrator system is described in Grefe and Kaiser (2014). Briefly, a percolating glass bed equilibrator was connected to a $N_2O/CO$ analyser ($N_2O/CO$-23d, Los Gatos Research Inc.). Artificial air mixtures (21 % $O_2$, 79 % $N_2$) with nominal $N_2O$ mole fractions of 300, 320 and 340 nmol mol$^{-1}$ (BOC) were used as
reference gases. These gas mixtures were compared with IMECC/NOAA standards to determine the exact values of (297.6±0.1), (325.3±0.1) and (344.2±0.1) nmol mol$^{-1}$ (NOAA-2006 scale). Reference gases and marine background air were





measured twice a day for 20 min each during JR260B. Due to sufficient analyser stability during this cruise, calibration was reduced to once a day during JR255A/GENTOO. To ensure complete flushing of the cavity, only the last 5 min of each 20-min gas measurement were evaluated. Correspondingly, the first 15 min of equilibrator data following the reference gas and air measurements were discarded. Precision for single reference measurements (standard deviation of measurements at 1Hz

over 5 min) was 0.4 nmol mol$^{-1}$. The standard deviation of the uncorrected reference gas measurements throughout JR260B was 1.1 nmol mol$^{-1}$ ($n = 29$) and 0.8 nmol mol$^{-1}$ ($n = 19$) for JR255A/GENTOO. To estimate the long-term repeatability of the measurements after calibration, we used the standards with the lowest and highest N$_2$O mole fraction to calibrate the standard with the middle N$_2$O mole fraction. The mean difference between resulting calibrated N$_2$O mole fraction and the actual value of 325.3 nmol mol$^{-1}$ was (0.2±0.1) nmol mol$^{-1}$, corresponding to a precision better than 0.1 %. The flow rate of

the headspace gas through the analyser was 400 mL min$^{-1}$ (293 K, 1 bar) resulting in a 95 % relaxation time ($t_{95} = 3\tau$) of approximately 7 minutes (Grefe and Kaiser, 2014). Data was acquired at a rate of 1 Hz; which was binned into 60 s averages for data evaluation purposes. Water flow through the equilibrator was held constant at 1.8-1.9 L min$^{-1}$, using a flow regulator (Robert Pearson & Company Ltd, ½ inch diameter tap tail flow regulator). Water temperature in the equilibrator was measured with a Pt-100 temperature probe (Omega Engineering Limited), calibrated against a mercury reference

thermometer to within ±0.06 ºC, and recorded using a RTD temperature recorder and USB Datalogger interface (both Omega Engineering Limited).

## 3 Results and discussion

### 3.1 N$_2$O concentrations and saturations in the surface ocean

Concentration of dissolved N$_2$O ($c$) was calculated from the dry mole fraction measured in the equilibrator headspace ($x$),

water temperature in the equilibrator ($T_{eq}$), salinity ($S$) and atmospheric pressure ($p_{air}$) using the solubility function $F$ as described by Weiss and Price (1980), Eq. (1).

$$c = xF(T_{eq}, S)p_{air} ,$$                                                                                (1)

N$_2$O saturation in surface waters ($s$) was calculated by comparing $x$ with the atmospheric mole fraction $x_{air}$ and the respective equilibrium concentrations for ($T_{eq}$) and temperature at the seawater intake ($T_{in}$) Eq. (2).

$$s = \frac{xF(T_{eq}, S)}{x_{air}F(T_{in}, S)}$$                                                          (2)

Values for $p_{air}$, S and $T_{in}$ were from the ship's surface water and meteorological monitoring system Surfmet (http://www.bodc.ac.uk).

N$_2$O dry mole fractions measured in marine air were on average (323.6±0.6) nmol mol$^{-1}$ during JR260B and (324.0±0.7) during JR255A/GENTOO. These values agree within measurement uncertainties with data from the Advanced Global



Atmospheric Gases Experiment (AGAGE) for Cape Grim, Tasmania in January 2012 (40.68° S 144.69° E, (323.9±0.5) nmol mol$^{-1}$).

### 3.1.1 JR160B

Concentrations and saturations of $N_2O$ in the surface ocean for JR260B are shown in Fig. 2a and b, respectively. The average

$N_2O$ concentration was (14.0±0.7) nmol L$^{-1}$, with lowest values of 11.2 nmol L$^{-1}$ near the Falkland Islands and highest values of 16.2 nmol L$^{-1}$ while crossing the Polar Frontal Zone (PFZ) during transit to South Georgia.

The average $N_2O$ saturation was (104.3±3.4) %. Lowest saturations of 99.4 % were observed in the Subantarctic Zone (SAZ) during transit from South Georgia back to the Falkland Islands. Saturations were highest to the east of the PF (126.1 %) where surface concentrations were also highest.

*Subantarctic Zone SAZ*

In the Subantarctic Surface Waters (SASW) close to the Falkland Islands to the east of the Subantarctic Front (SAF), average surface concentrations were low, while saturations were similar to those observed in the Antarctic Zone (AAZ) (11.4±0.2) nmol L$^{-1}$, (102.4±0.7) %, see below). Water temperature and salinity were higher than the Antarctic waters south and east of the PF, so the difference in concentrations was likely due to solubility effects. The Falkland current transports nutrient-rich

waters from the ACC onto the Falkland shelf (Peterson and Whitworth, 1989), potentially supporting $N_2O$ production during remineralisation in the SAZ.

*Polar Frontal Zone PFZ*

$N_2O$ concentrations across the PFZ between the SAF and PF were highly variable with values ranging from 11.4 to 16.2 nmol L$^{-1}$ (101.7 to 126.1 % saturation) while water temperature and salinity were decreasing towards the east. Average

concentrations and saturations across the PFZ were (13.8±0.9) nmol L$^{-1}$ and (108.8±5.2) %, respectively.

The most important sources of $N_2O$ to surface waters are upwelling of old water masses with preformed high $N_2O$ concentrations and in situ production during remineralisation or denitrification. High $N_2O$ saturations across the PFZ could neither be attributed to a specific water mass, nor to a point on the mixing line between Subantarctic Surface Water (SASW) and Antarctic Surface Water (AASW) (Fig. 3).

Frontal systems can supply nutrients and iron to surface waters with large phytoplankton blooms forming across the PFZ, potentially fuelled by iron input from the Antarctic Peninsula archipelago, the Scotia Ridge and Georgia Rise (de Baar et al., 1995). Re-mineralisation of sinking bloom-biomass could lead to enhanced in situ $N_2O$ production across the PFZ, resulting in the high saturations values observed during JR260B.

*Antarctic Zone AAZ*



Average $N_2O$ concentrations in the Antarctic Zone (AAZ) to the west of the PF near South Georgia were substantially higher than in the SAZ (14.1±0.4) nmol $L^{-1}$. Saturations, however, were only slightly increased ((103.9±3.1) %), similar to values of (103.0±2.0)% previously observed in the region (Weiss et al., 1992). The lower water temperature and salinity of Antarctic Surface Water AASW increased $N_2O$ solubility, resulting in lower saturation values. Upwelling of deep water masses would

not be expected away from the ACC and denitrification is not likely to take place in the oxygenated surface waters of the AAZ. Nitrification and nitrifier-denitrification would be more likely sources of in situ $N_2O$ production. High nitrification rates (> 30 mmol $m^{-2}$ $d^{-1}$) were suggested for the Pacific sector of the Southern Ocean south of the PF (Sambrotto and Mace, 2000) and could also account for $N_2O$ supersaturation observed in the AAZ of the Atlantic sector. Extensive phytoplankton blooms were observed in the vicinity of South Georgia and across the Scotia Sea, extending to the southern limit of the Polar

Front (Korb et al., 2004, 2005). These blooms can develop due to a stable water column over the shelf and iron input from the shelf sediments and island runoff (de Baar et al., 1995; Holeton et al., 2005; Korb et al., 2005; Wadley et al., 2014). This accumulation of biomass would supply ample substrate to sustain high nitrification rates and $N_2O$ production could lead to the observed supersaturating close to the island. Not much is known about nitrifier-denitrification in polar waters, additional research is required to estimate the importance of this production pathway for $N_2O$ accumulations in surface waters.

*Stromness Bay*

$N_2O$ concentrations and saturations in coastal Stromness Bay, South Georgia, were higher than in the open waters of the AAZ ((14.8±0.3) nmol $L^{-1}$ and (108.1±2.6) %, respectively) with highest saturations observed where fresh meltwater runoff from the island was mixing with AASW (Fig. 3). Terrestrial runoff can transport iron and biomass from land into the sea and stimulate productivity and subsequent remineralisation where $N_2O$ is produced during nitrification. In addition, fur seals

(*Arctocephalus gazella*) and macaroni penguins (*Eudyptes chrysolophus*) have large breeding colonies on South Georgia, re-distributing nitrogen from their hunting grounds to the island (Whitehouse et al., 1999). The high nitrogen load in coastal waters is expected to lead to high $N_2O$ saturations (Bange et al., 1996). Nitrifier-denitrification in anoxic sites of suspended particles can be an additional source of $N_2O$ (Ostrom et al., 2000). As the water depth in Stromness Bay is shallow (60 m) and well mixed at the anchoring site, $N_2O$ produced by denitrification at the sediment interface could have diffused into the

water column, contributing to the high concentration at the surface.

### 3.1.2 JR255A/GENTOO

During JR255A/GENTOO, average $N_2O$ surface concentrations were (14.9±1.2) nmol $L^{-1}$, corresponding to saturations of (103.1±3.6) %. While $N_2O$ concentrations were higher than for JR260B, mainly due to the lower water temperatures, average saturations were slightly lower. Lowest $N_2O$ surface concentrations of 10.6 nmol $L^{-1}$ were observed on the South

American shelf close to the Falkland Islands (Fig. 4a). Concentrations and saturations were highest across the South Scotia Ridge (up to 17.0 nmol $L^{-1}$, corresponding to 116.0 % saturation, Fig. 4b). The lowest saturations were observed close to the sea ice edge in the open waters of the Weddell Sea (96.5 %).



*Drake Passage*

Average N$_2$O concentrations at the beginning of the cruise were (13.7±1.8) nmol L$^{-1}$ ((102.9±4.2) % saturation). Decreasing temperatures during transit across Drake Passage to the Eastern Antarctic Peninsula partially accounted for increasing N$_2$O concentrations (Fig. 4a), resulting in a wide range of concentration values across the region, whereas saturations were on

average only slightly above equilibrium with the atmosphere. Water masses across Drake Passage were Subantarctic Surface Water to the north of the SAF and Antarctic Surface Water to the south of the PF (SASW and AASW, respectively) (Fig. 5). These surface waters are in constant contact with the atmosphere and are expected to be in equilibrium with the atmosphere in absence of biological N$_2$O sources. N$_2$O saturations above 100 % could be due to in situ production during remineralisation of biomass. For comparison, Rees et al. (1997) observed slightly lower saturations for Drake Passage

((99.7±3.0)%), while Weiss et al. (1992) reported values close to the data presented here (Ajax 2: (102.3±0.9)%, JR255A/GENTOO: (102.9±4.2)%). Higher saturation values for Weiss et al. (1992) and JR255A/GENTOO could be due to interannual and seasonal variability. JR255A/GENTOO and Ajax 2 took place in January and February, later in the austral summer season than the November/December cruise of Rees et al. (1997). Nitrification as part of the remineralisation of sinking biomass could have increased N$_2$O accumulations in the surface over the summer, resulting in higher values later in

the growing season.

Just south of the Southern Boundary of the Antarctic Circumpolar Current (SB), concentrations, as well as saturations, reached the highest values observed during JR255A/GENTOO. Temperature and salinity characteristics of the surface water indicated an influence of Upper Circumpolar Deep Water (UCDW) from below (Fig. 5, Orsi et al., 1995). UCDW is an old water mass, high in remineralised nutrients and N$_2$O as a byproduct of nitrification. Iron input from the Scotia Ridge, as

observed by Klunder et al. (2014) could additionally enhance productivity, supplying substrate in form of sinking particles for in situ N$_2$O production.

*Weddell Sea*

Surface N$_2$O concentrations of the open waters of the Weddell Sea were on average (15.2±0.2) nmol L$^{-1}$ ((102.2±1.6) % saturation). South of the ACC, the surface waters were colder than across Drake Passage, increasing the solubility of N$_2$O

and resulting in slightly lower saturations despite higher concentrations.

Higher-than-average N$_2$O saturations were observed on the Antarctic shelf off the tip of Joinville Island ((104.2±1.7) %) and during the section across the large standing eddy forming over the South Scotia Ridge, centred on 62° S and 54° W (Thompson et al., 2009) ((110.5±1.2) %) (Fig. 4a and b). Iron from the sediments, as well as land run-off could have stimulated productivity in these areas (Klunder et al., 2014; Sañudo-Wilhelmy et al., 2002). In addition, the sampling region

was more sheltered from the circumpolar winds by the Antarctic Peninsula, stabilising the water column over the shallow bathymetry. The combination of ample supply of trace nutrients and a stable water column, keeping phototrophic organisms



in the euphotic zone, presumably resulted in comparably high productivity and subsequent $N_2O$ production during remineralisation of sinking biomass.

*Sea ice edge*

Lowest saturations of on average (98.8.0±1.3) % were observed close to the sea ice edge in the south east of the survey region (Fig. 4b). $N_2O$ concentrations were similar to other open ocean regions in the Weddell Sea ((15.4±0.2) nmol L$^{-1}$ at the ice edge compared to (15.2±0.2) nmol L$^{-1}$). Saturations, however, were lower due to lower temperature and salinity, resulting in higher $N_2O$ solubility. Randall et al. (2012) observed under-saturations of $N_2O$ within sea ice due to loss of dissolved gases during brine rejection. Mixing of seawater with under-saturated melt water would decrease surface concentrations, which was not observed during JR255A. However, it is possible that end-members were not captured in these measurements and the mixing line for temperature and salinity indeed indicates dilution of Weddell Sea surface waters with melt water low in $N_2O$.

### 3.2 $N_2O$ sea-to-air flux

Sea-to-air flux was calculated using wind speeds from the CCMP Wind Vector Analysis Product (www.remss.com/measurements/ccmp). The gas transfer coefficient ($k_w$) was calculated, using the parameterisation of Nightingale (2000) (Equation 3) and $k_w$ was adjusted for $N_2O$ with the Schmidt number *Sc* calculated following (Wanninkhof, 1992), Eq. (3).

$$\frac{k_w}{\text{m d}^{-1}} = 0.24 \left[ 0.222 \left( \frac{u_{CCMP}}{\text{m s}^{-1}} \right)^2 + 0.333 \frac{u_{CCMP}}{\text{m s}^{-1}} \right] \left( \frac{Sc}{600} \right)^{-0.5} \tag{3}$$

The air-sea flux ($\Phi$) was calculated from $k_w$ and the difference between $N_2O$ concentrations in seawater $c$ and air saturation concentrations ($c_{sat}$), Eq. (4):

$$\Phi = k_w (c - c_{air}) = k_w \left[ c - x_{air} \frac{u_{CCMP}}{\text{m s}^{-1}} \right] \left( \frac{Sc}{600} \right)^{-0.5} \tag{4}$$

### 3.2.1 JR160B

Surface waters during JR260B were mainly a source of $N_2O$ to the atmosphere (Fig. 6). The average flux of 2.3 µmol m$^{-2}$ d$^{-1}$ was higher than the global average flux of 1.1 µmol m$^{-2}$ d$^{-1}$ (based on a marine contribution of 25 % to global $N_2O$ emissions (Ciais et al., 2013)).

The average sea-to-air flux within the SAZ was ((1.1±0.4) µmol m$^{-2}$ d$^{-1}$), due to low to moderate wind speeds at the time of measurements and saturations just slightly above equilibrium with the atmosphere.

$N_2O$ flux was increasing across the PFZ to an average of (3.6±2.4) µmol m$^{-2}$ d$^{-1}$ with highest values of 13.3 µmol m$^{-2}$ d$^{-1}$ during transit from South Georgia back to the Falkland Islands. Average global flux values were highly exceeded across the



frontal zone as high $N_2O$ supersaturations, possibly a result of in situ production due to high biomass supported by iron supply from sediments, coincide with high circumpolar wind speeds.

Sea-to-air flux decreased again in the open waters of the AAZ around South Georgia as $N_2O$ supersaturations were lower. The average flux of (1.8±1.8) µmol m$^{-2}$ d$^{-1}$ was, however, still exceeding the global average. Negative flux up to -0.5 µmol

m$^{-2}$ d$^{-1}$ was only observed briefly to the east of the PF during transit to South Georgia.

In contrast to the open waters of the AAZ, Stromness Bay was a strong source of $N_2O$ to the atmosphere while the ship was anchored for calibration of acoustic instruments. Sea-to-air flux was initially only slightly higher than for the surrounding open ocean area (2.2 µmol m$^{-2}$ d$^{-1}$), despite high $N_2O$ supersaturations in the bay. This was due to low wind speeds of 7.3 m s$^{-1}$, which subsequently increased to 14.5 m s$^{-1}$ throughout the day, resulting in sea-to-air fluxes of up to 10.6 µmol m$^{-2}$ d$^{-1}$

and an average of (4.5±2.4) µmol m$^{-2}$ d$^{-1}$.

Overall, the surface ocean was a strong source of $N_2O$ to the atmosphere for most of the research cruise JR260B with sea-to-air fluxes exceeding global average flux. These high flux rates were driven by high surface water saturations, presumably resulting from in situ production, and moderate to high wind speed.

### 3.2.2 JR255A/GENTOO

The average sea-to-air $N_2O$ flux throughout JR255A/GENTOO was (0.7±0.9) µmol m$^{-2}$ d$^{-1}$, which is below global average values of 1.1 µmol m$^{-2}$ d$^{-1}$ and considerably lower than the average flux for JR260B, just to the north of the JR255A/GENTOO region. The low flux was a result of lower saturation values compared with JR260B, but still above equilibrium with the atmosphere for most of the cruise. Additionally, wind speeds were lower during JR255A/GENTOO.

Highest sea-to-air flux of 6.8 µmol m$^{-2}$ d$^{-1}$ was observed across the South Scotia Ridge where high wind speeds coincided

with high $N_2O$ supersaturations, presumably due to surface waters being influenced by underlying UCDW high in $N_2O$.

Although surface waters were most strongly undersaturated at the sea ice edge, negative fluxes were highest at the beginning of the cruise, close to the Falkland Islands (-1.3 µmol m$^{-2}$ d$^{-1}$). Highest wind speeds of up to 13.3. m s$^{-1}$ were observed at the beginning of the cruise, driving the strong negative fluxes on the shelf. At the ice edge, in contrast, wind speeds were considerably lower (1.4 to 6 m s$^{-1}$), resulting in weaker negative fluxes.

Low wind speeds also affected sea-to-air flux on the Antarctic shelf close to Joinville Island, as well as across the standing eddy over the South Scotia Ridge. Despite relatively high supersaturations, $N_2O$ flux was rather low ((0.5±0.2) and (0.5±0.4) µmol m$^{-2}$ d$^{-1}$, respectively).

Throughout JR255A sea-to-air flux of $N_2O$ was low compared to JR260B and the global average, mainly due to low wind speed rather than low $N_2O$ saturation of surface waters. Areas like the continental shelf, eddies and the ice edge hold the

potential for substantial $N_2O$ sources and sinks if wind speed increases and production in the water column is sustained.

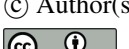



**Summary**

For both cruises, JR260B and JR255A/GENTOO, high $N_2O$ saturations were observed over shallow bathymetry across the North and South Scotia Ridge and on the Antarctic shelf and coastal South Georgia. This might be due to iron input supporting biomass production, which then forms the substrate for $N_2O$ production (Fuhrman and Capone, 1991; Jin and Gruber, 2003).

Frontal systems represented another source region for $N_2O$. To the east of Drake Passage during JR260B, high $N_2O$ saturations across the PFZ were presumably the result of in situ production during remineralisation of sinking biomass. South of Drake Passage across the South Scotia Ridge/south of the SB during JR255A UCDW influencing surface waters resulted in a strong source of $N_2O$ to the atmosphere. The frontal system of the ACC can be an important source of $N_2O$, but the observed supersaturation was highly variable. High-resolution measurements are a valuable tool for accurately assessing $N_2O$ emissions from this region.

Assuming that the average flux values of 2.3 $\mu mol\ m^{-2}\ d^{-1}$ and 0.7 $\mu mol\ m^{-2}\ d^{-1}$ calculated for JR260B and JR255A are representative for the Scotia Sea and Weddell Sea over time, the combined area could contribute 0.04 Tg $a^{-1}$ N (nitrogen equivalent) to the global $N_2O$ source. While this value is relatively low, also compared to model estimates of 0.9 Tg $a^{-1}$ N (nitrogen equivalents) for the entire Southern Ocean (Nevison et al., 2005), areas with a strong variability in $N_2O$ saturation and sea-to-air flux were observed for both cruises. These findings indicate the importance of high-resolution data to accurately estimate the source strength of mesoscale features, such as frontal systems and eddies.

*Competing interests*: The authors declare that they have no conflict of interest.

**Acknowledgements**

We would like to thank the officers and crew on board RRS James Clark Ross and the scientific party for their support during JR260B and JR255A/GENTOO. We also thank Sunke Schmidtko for supplying calibrated sea surface temperature and salinity, Dorothee Bakker for providing the equilibrator used in this study and Gareth A. Lee for technical support. This study was supported by the European Community's Seventh Framework Programme (FP7/2007-2013) under grant agreement number 237890 (Marie Curie Initial Training Network "INTRAMIF") and the NERC Collaborative Gearing Scheme, project number AFI CGS78. We acknowledge the BAS Ecosystems Long Term Monitoring and Survey Programme Western Core Box (LTMS WCB).

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





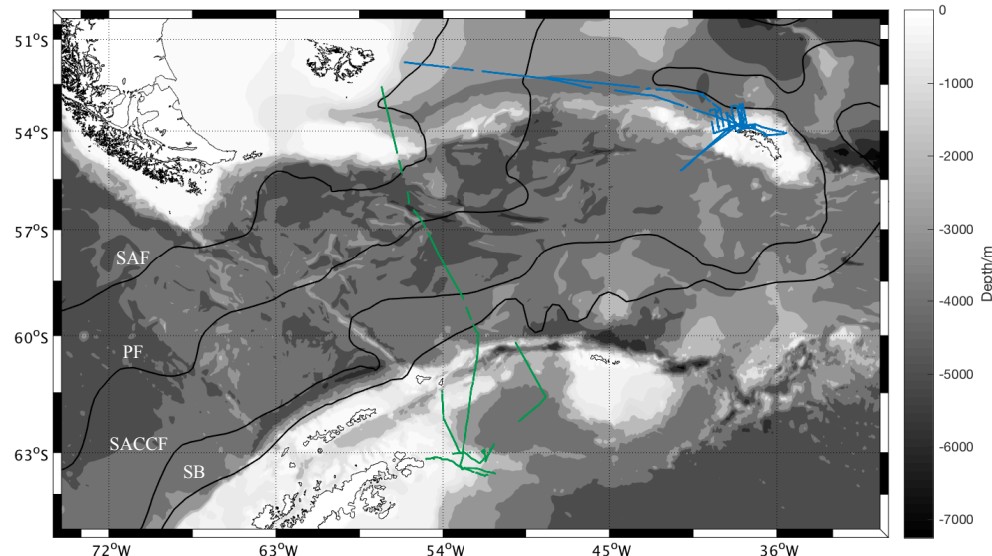

**Figure 1:** Sampling area with bathymetry from the General Bathymetric Chart of the Oceans (GEBCO) one minute grid. Isobaths every 1000 m between 8000 m and 1000 m depth, every 200 m between 1000 m and 0 m. N$_2$O measurement positions during JR260B in blue, N$_2$O measurement positions during JR255A/GENTOO in green. Climatological locations of fronts in black after Orsi et al. (1995).

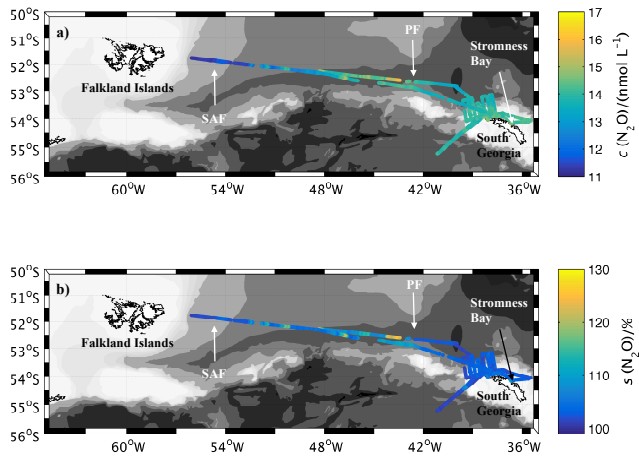




**Figure 2:** a) $N_2O$ concentration in surface waters along the cruise track of JR260B. b) $N_2O$ saturations calculated from measured atmospheric and seawater dry mole fractions. Approximate position of Subantarctic Front (SAF) and Polar Front (PF) indicated, based on sea surface temperature and salinity measurements during JR260B.

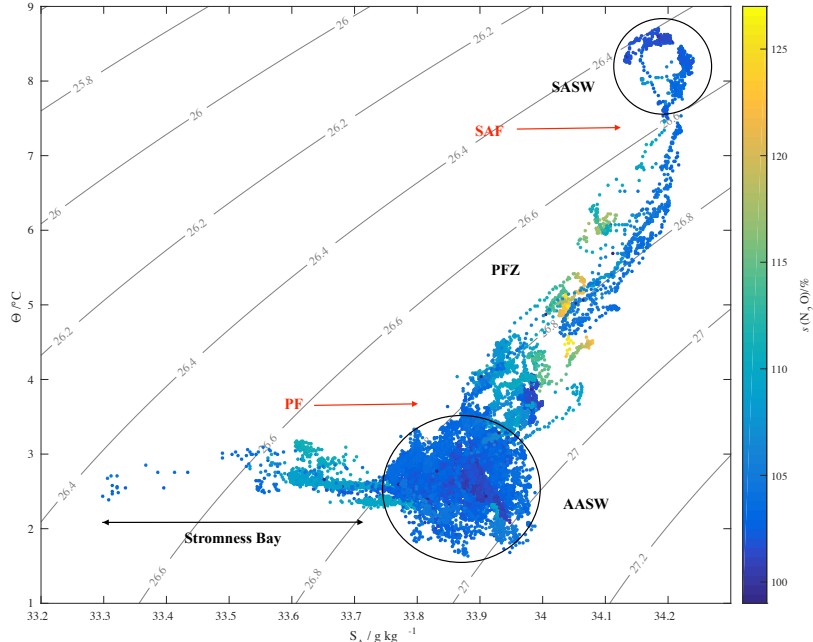

5    **Figure 3:** $N_2O$ saturations $s(N_2O)$ with corresponding absolute salinities $S_A$ and conservative temperatures $\Theta$ for JR260B. Contour lines are the potential density anomalies ($\sigma_\theta$) calculated using the TEOS-10 GSW toolbox. Approximate positions of the Subantarctic Front (SAF) and Polar Front (PF) based on sea surface temperature and salinity measurements during JR260B are indicated by red arrows. SASW: Subantarctic Surface Water, AASW: Antarctic Surface Water.



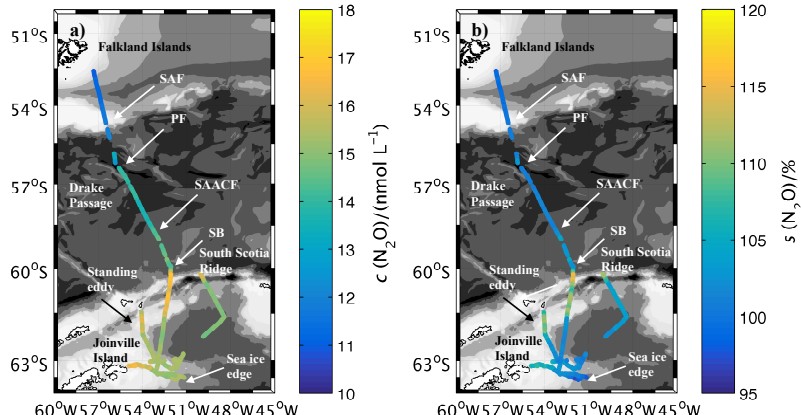

**Figure 4:** a) N$_2$O concentrations in surface waters during JR255A/GENTOO. b) Saturations, calculated measured atmospheric and seawater dry mole fractions. Approximate positions of the Subantarctic Front (SAF), Polar Front (PF), Southern Antarctic Circumpolar Current Front (SAACF) and Southern Boundary of the Antarctic Circumpolar Current (SB) based on sea surface temperature and salinity measurements during JR255A.

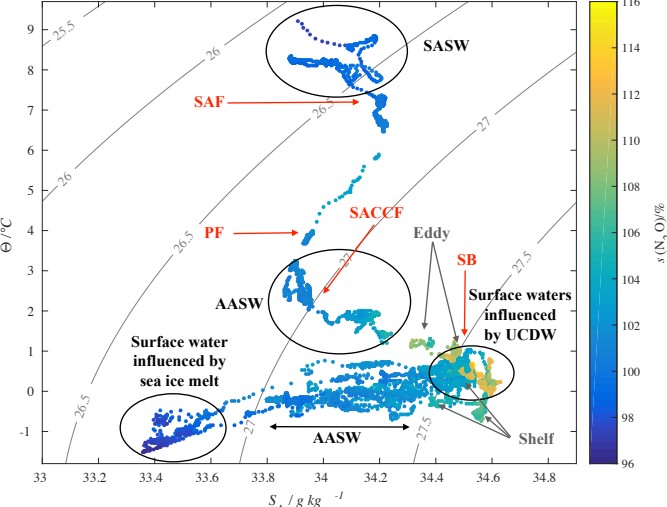

**Figure 5:** N$_2$O saturations $s$(N$_2$O) with corresponding absolute salinities S$_A$ and conservative temperatures Θ for JR255A/GENTOO. Contour lines are the potential density anomalies calculated using the TEOS-10 GSW toolbox. Approximate positions of the Subantarctic Front (SAF), Polar Front (PF), Southern Antarctic Circumpolar Front (SACCF) and Southern Boundary of the Antarctic Circumpolar



Current (SB) based on sea surface temperature and salinity measurements during JR255A are indicated by red arrows. SASW: Subantarctic Surface Water, AASW: Antarctic Surface Water, UCDW: Upper Circumpolar Deep Water.

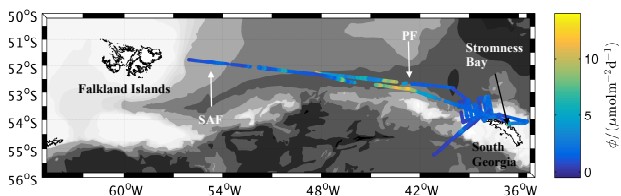

**Figure 6:** Sea-to-air flux along the cruise track of JR260B. Approximate positions of the Subantarctic Front (SAF) and Polar Front (PF) based on sea surface temperature and salinity measurements during JR260B.

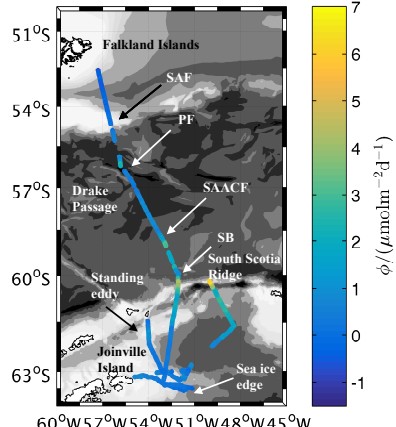

**Figure 7:** Sea-to-air flux along the cruise track of JR255A. Approximate positions of the Subantarctic Front (SAF), Polar Front (PF), Southern Antarctic Circumpolar Front (SACCF) and Southern Boundary of the Antarctic Circumpolar Current (SB) based on sea surface temperature and salinity measurements during JR255A.