# Peer review of "Nitrous oxide variability at sub-kilometre resolution in the Atlantic sector of the Southern Ocean"

_Biogeosciences, 2017_

## Author Comment (AC1) · 13 Mar 2017

Equation 4 in the discussion paper is incorrect; the equation should read:

$$\Phi = k_{\mathrm{w}}(c - c_{\mathrm{air}}) = k_{\mathrm{w}}[c - x_{\mathrm{air}}F(T_{\mathrm{in}}, S)p_{\mathrm{air}}]$$

---

## Referee Comment (RC1) · Anonymous Referee #1 · 15 Mar 2017

The manuscript by Grefe and coauthors describes nitrous oxide concentrations (and thereby saturation and air-sea exchange) in the Southern Ocean. Data is provided from underway measurements conducted during a 18 day and a 14 day expedition. The dataset serves as a reminder that nitrous oxide can be both super-saturated and under-saturated at high latitudes and demonstrates that underway measurements allow concentrations to be attributed to physical features such as frontal systems which presumably have increased biomass and microbial activity. The manuscript is well-written, well-presented, and builds off a previous methodological paper by the same authors.

The biggest drawback to this manuscript is the short length and lack of novelty. The stated objective of the manuscript is to address the lack of nitrous oxide measurements in the Southern Ocean, yet only 14 days of measurements in the Southern

Ocean (cruise ID: JR255A) are presented. It's not clear to me how much of the 18 day expedition (cruise ID: JR260B) which went from the Falkland Islands to S Georgia took place in the Southern Ocean which as I understand is defined as the area south of the Antarctic Convergence. I realize the importance of field data and believe these measurements to have been conducted to a high standard, but with limited data the interpretation is limited and the reader is left wondering what new information or insight did I learn from reading this article.

I recommend this article be considered as a technical note or combined with other datasets for a more substantial contribution.

---

## Referee Comment (RC2) · Anonymous Referee #2 · 26 Apr 2017

The manuscript by Grefe et al. presents the results of continuous, high-resolution surface measurements of N2O in the Southern Ocean (SO). The authors use measured concentrations as well as computed saturations and sea-to-air fluxes in order to describe the high spatial variability that can be found across the major fronts and zones in the Atlantic sector of the SO. Likewise, the authors discuss potential physical/microbial mechanisms which could explain the observed distribution of N2O and the magnitude of the concentration/saturation/sea-to-air fluxes differences across the areas covered by two expeditions between December 2011 and February 2012. The manuscript is well written and structured and it further illustrates the advantages of using spectroscopic methods for large-scale surveys of trace gases such as N2O.

General comments

Although I see no problem with the length of the manuscript per se, I think several aspects need to be addressed before considering its publication. I have particular concerns with respect to the repeated reference to N2O production processes at depth in the absence of depth profiles of N2O or any other parameters which could support that argument. Although the potential mechanisms discussed by Grefe et al. are reasonable, one cannot avoid wondering at what extent they do affect the distribution of N2O in comparison with physical processes (such as upwelled waters in e.g. the Antarctic Zone just south of the Polar Front) and/or solubility effects. Since a great deal of attention of given to primary production as a potential source for enhanced organic material which upon sinking could fuel N-cycling in the water column, I would expect to see at least remote sensing data in order to "back up" the interpretation of the N2O distribution. If no ancillary data from the JR260B and JR255A cruises is available, one could think about climatological products of oxygen and nutrient distributions (e.g. from the WOA data). There is also a recent study by Rees et al. (Deep Sea Res., 127, 2016) in the Scotia Sea. Although no depth profiles of N2O were done during their SO campaign, they show background oceanographic conditions for this area in January-February 2013 as well as a discussion on N2O production pathways which could be potentially useful for this study.

I believe that this study is based in high-quality data. However I also think its relevance could come across more clearly if the authors would set more focus in what the sea-to-air fluxes mean for the oceanic budget of N2O, rather than in its production pathways in the water column since there seem to be little data on that. As it stands now, the discussion on emissions of N2O out of the SO is relegated to a few lines in sections 3.2.1, 3.2.2 and the summary. I suggest the authors to further elaborate on the reported emissions and its comparison to the global budget. This would mean, among others, to include a detailed description of how these estimates were obtained because this information cannot be inferred from the current manuscript. Also, I kindly suggest the authors to consider including a "Conclusions" section in which the main points of the study are clearly stated, since this would help the reader to better understand its

importance.

Likewise, I think the manuscript might benefit from a comparison between the data presented and data from other sectors of the SO and at comparable latitudes (see e.g. Zhan and Chen, JGR 114, 2009; Chen et al., Acta Oceanologica Sinica 33(6), 2014; Farías et al., BGS 12, 2015; Zhan et al., JGR: Oceans 120(8), 2015). Such a comparison might stress the point on why is it needed to start looking at the SO, and in particular the Atlantic sector for which, as the authors correctly point out, data coverage is scarce.

One more aspect which would further help the reader throughout the paper is to have a brief, yet substantial introduction of the study area, and in particular of the zonal circulation since it is used extensively to explain the surface N2O variability between the different fronts and zones occupied during the cruises.

Specific comments

P.1 L.22: I guess the authors meant to cite Ciais et al. (2013) instead of Myhre et al., (2013), since Myhre et al. deals with radiative forcing of GHGs rather than with their budgets and reservoirs.

P.1 L.26: In this context it should be Nevison et al. (2005) and not (2004).

P.2 L. 20-21: "capture the impact of frontal structures and changes in weather conditions on emissions from the Southern Ocean." If this appear as one of the major goals of the manuscript, the discussion on this topic should be extended (see comment above)

P.3 L.22: Remove comma after Eq. 1.

P.4 L.3: It should be JR260B not JR160B.

P.4 L.4-9: This paragraph is redundant with the detailed descriptions given on the following sections. I suggest the authors to modify this part to include a more general

statement that introduces the individual results for each area.

P.4 L.8: The authors observed the lowest N2O saturations in the SAZ when approaching the Falkland Islands. Yet, in P.4 L.14-16 they also state that nutrient rich waters might potentially stimulate N2O production in the SAZ. What could be then the reason for the low N2O saturations? Please clarify.

P.4 L.22-24: "High N2O saturations across the PFZ could neither be attributed to a specific water mass, nor to a point on the mixing line between Subantarctic Surface Water (SASW) and Antarctic Surface Water (AASW) (Fig. 3).". The meaning of this sentence is not completely clear, please check.

P.4 L.27: It should be Remineralisation.

P.4 L.27: "bloom-biomass". Hyphenating here seems incorrect to me. Please check.

P.5 L.13: It should be supersaturation.

P.5 L.20-25: Again, although this argument seems feasible, in this case it remains highly speculative since no data to substantiate it is presented. If this data do exist, it should be included or explained in more detail here.

P.6 L.5: (. . .) across the Drake Passage (. . .)

P.6 L.7-8: "These surface waters are in constant contact with the atmosphere and are expected to be in equilibrium with the atmosphere in absence of biological N2O sources.". This sentence is redundant, please rephrase.

P.6 L.13-15: "Nitrification as part of the remineralisation of sinking biomass could have increased N2O accumulations in the surface over the summer, resulting in higher values later in the growing season.". What is the definition of surface in this context? In this region the ML can be quite large and therefore I could imagine the "surface layers" being deep enough to hold a nitrifying layer from which N2O can be mixed into the upper layers. If on the other hand "surface" is meant in the sense of the depth at

which the measurements are collected (I would guess ca. 6-7 m), I would find it hard to use nitrification as an explanation since I would not expect this process to occur at important rates in well-oxygenated waters in close contact with the atmosphere. Please clarify.

P.6 L.18-19:" UCDW is an old water mass, high in remineralised nutrients and N2O as a byproduct of nitrification.". Are there any studies showing enhanced N2O in association with the UCDW or was this rather inferred from the biogeochemical properties of this water mass? Please clarify or cite the corresponding sources.

P.6 L.26: What does "Higher-than-average" means in this context? With respect to the global open ocean mean? With respect to the observations in this study? Please clarify.

P.7 L.5-6: "N2O concentrations were similar to other open ocean regions in the Weddell Sea ((15.4±5 0.2) nmol L-1 at the ice edge compared to (15.2±0.2) nmol L-1)". This sentence is not clear, please reorganize.

P.7 L.12-20: This whole paragraph, including Eqs. 3 and 4, would fit better in the methods section.

P.7 L.21: It should be JR260B not JR160B.

P.7 L.22-24: Although the IPCC is a good reference for global budgets, I would also compare with the observations-based estimate from Nevison et al. (1995).

P.7 L.27: Suggestion: "N2O flux progressively increased across (. . .)".

P.8 L.8-10: Did the N2O saturation values remained stable during the period for which the increased wind speeds were observed? It is clear that for a given saturation higher wind speeds would yield higher fluxes but if there were high changes in saturation this might indicate other processes such as transport of air masses from land, changes in water properties, etc. Also it would be useful for the interpretation to clearly state the time scale of this change; was it a 24 h cycle? Longer?

P.8 L.28: It should be JR255A/GENTOO.

P.8 L.29: Eddies are not an area but rather an oceanographic feature.

P.8 L.29-30: If N2O production is sustained at depth and then due to e.g. wind-driven mixing this N2O is transported to the surface, and ultimately to the atmosphere via air-sea gas exchange, I would say this is only a source, not a sink as it is stated here.

P.9 L.3-5: I would recommend the authors to exclude citations from the summary since they are more suited for the results and discussion section.

References

There are a few misspellings, please check thoroughly.

Figures

Figure 1: Considering the importance of the fronts for this study I think this figure should have more emphasis in e.g. SST or any equivalent indicator of the position of the fronts and zones of the study area, and less in the bathymetry. Also the cruise tracks are quite difficult to visualize on the current figure. One further recommendation is to highlight better the location of the fronts, zones and the different locations which are discussed on the manuscript. As a reader, I find this would make much easier to follow the description of the results. Also, since the GEBCO data base was used to produce part of this figure, it could be that the use of this data needs to be acknowledged in the list of references, please check.

Figure 2: I would enlarge the figures in order to appreciate better the fine structure of the distribution of N2O concentrations/saturations.

Figure 3: Please increase the font size of this plot. May be it is worth mentioning the usage of the TEOS-10 toolbox in the methods section instead of in the figure caption.

Figure 4: Same comment as for Figure 2 regarding the figure size. Moreover, in the caption it should read: "b) N2O saturations, calculated from measured (. . .)"

Figure 5: Same comments as for Figure 3.

Figure 6-7: I would suggest the authors to consider omitting these two plots from the manuscript since they do not provide additional information. If the authors want to show areas with relatively low N2O saturations and still high fluxes due to enhanced wind speeds, it would be more useful to show a time series of wind speeds, saturations and fluxes in which such locations can be highlighted.